# Testing the Effect of Sampling Effort on Inferring Phylogeographic History in *Psolodesmus mandarinus* (Calopterygidae, Odonata)

**Liang-Jong Wang** [1,*]**, Yen-Wei Chou** [1] **and Jen-Pan Huang** [2,*]

1   Division of Forest Protection, Taiwan Forestry Research Institute, Taipei 10066, Taiwan
2   Biodiversity Research Center, Academia Sinica, Taipei 11529, Taiwan
*   Correspondence: ljwang23@ms17.hinet.net (L.-J.W.); airbugshuang@gmail.com (J.-P.H.)

**Abstract:** Phylogeographic studies have revealed spatial genetic structure and inferred geographical processes that may have generated genetic diversity and divergence. These study results have implications not only on the processes that generate intraspecific and interspecific diversity but also on the essential integrals for defining evolutionary entities (e.g., species). However, the resulting phylogeographic inferences might be impacted by the sampling design, i.e., the number of individuals per population and the number of geographic populations studied. The effect of sampling bias on phylogeographic inferences remains poorly explored. With a comprehensive sampling design (including 186 samples from 56 localities), we studied the phylogeographic history of a Taiwanese endemic damselfly, *Psolodesmus mandarinus*, with a specific focus on testing the impact of the sampling design on phylogeographic inference. We found a significant difference in the genetic structure of eastern and western populations separated by the Central Mountain Range (CMR) of Taiwan. However, isolation by the CMR did not lead to reciprocally monophyletic geographic populations. We further showed that, when only a subset of individuals was randomly included in the study, monophyletic geographic populations were obtained. Furthermore, historical demographic expansion could become undetectable when only a subset of samples was used in the analyses. Our results demonstrate the impact of sampling design on phylogeographic inferences. Future studies need to be cautious when inferring the effect of isolation by a physical barrier.

**Keywords:** Zygoptera; molecular phylogeny; network; population genetics; sampling effect

## 1. Introduction

Different sampling efforts may impact phylogeographic inferences [1–4]. For example, coalescent simulations based on multilocus data have been shown to more accurately reconstruct population history than those resulting from the use of single locus datasets [5]. Additionally, sampling an insufficient number of individuals from each population/locality may also impact the estimated population genetic parameters (e.g., the genetic diversity parameter θ), which may lead to biased parameter values for simulation-based studies (e.g., approximate Bayesian computation, ABC, methods [1]) and therefore support erroneous phylogeographic histories. Because most of the conventional phylogeographic studies rely heavily on the inferred gene tree, particularly the mitochondrial gene tree [5], the sensitivity of phylogenetic reconstruction due to taxa sampling could have profound effects on the reconstructed phylogeographic history [6]. The importance of inferences from phylogeographic studies extend beyond semantic issues. For example, identifying areas of high genetic diversity, e.g., historical climatic refugia, and distinct genetic entities, e.g., cryptic species, can significantly influence conservation strategies [7]. However, rather than testing for insufficient sampling, many phylogeographic studies intrinsically assume that their sampling design represents the true distribution of genetic diversity across the geographic distribution of the studied organism.

The subtropical island of Taiwan accommodates high biodiversity (both species and genetic diversity [8–10]) and has been the focus of extensive phylogeographic studies in the past two decades because of its recent, yet drastic tectonic history, which may have generated high levels of intraspecific genetic diversity (e.g., [8–10]). One of the main topics has been the effect of the Central Mountain Range (CMR) on driving population subdivision. Specifically, three phylogeographic patterns have been identified across multiple different evolutionary lineages: (1) different geographic populations separated by mountain ranges form monophyletic lineages, indicating that the CMR (or mountain ranges in general) can effectively promote allopatric divergence; (2) significant genetic structure is found between geographic populations, which implies reduced gene flow because of the CMR; and (3) no significant geographic genetic structure (see Table 1 for a non-comprehensive summary). Although biological and ecological differences between organismal groups have often been argued to be responsible for the different phylogeographic patterns (e.g., freshwater associated species are often attributed to phylogeographic pattern 1 [11]; see Table 1), such differences in phylogeographic patterns could also result from differences in sampling effort. For example, when multiple molecular markers have been included in a study, different phylogeographic patterns have often been inferred (e.g., [9,11–14]). Additionally, studies that reveal insignificant geographic genetic structure (pattern 3) between eastern and western populations tend to include a smaller number of individuals (Table 1).

**Table 1.** Examples of Taiwanese phylogeographic studies and their evolutionary inferences.

| Organism | Sample Size | # Localities | Inferred Pattern * | Reference |
|---|---|---|---|---|
| Bamboo viper | 201 | 40 | 2 | [12] |
| Bat1 | 108 | 50 | 2 | [11] |
| Bat2 | 146 | 50 | 1 | [11] |
| Bat3 | 234 | 50 | 2 | [11] |
| Bat4 | 164 | 50 | 2 | [11] |
| Toad | 279 | 27 | 2 | [15] |
| Damselfly1 | 159 | 32 | 2 | [14] |
| Damselfly2 $ | 60 | 20 | 1 | [16] |
| Flying squirrel1 | 40 | 20 | 3 | [17] |
| Flying Squirrel2 | 35 | 18 | 3 | [17] |
| Freshwater Crab | 88 | 18 | 1 | [18] |
| Freshwater Prawn | 195 | 20 | 1 | [19] |
| Frog | 198 | 31 | 1 | [20] |
| Spider | 189 | 18 | 3 | [21] |
| Small mammal | 71 | 29 | 1 | [22] |
| Stag beetle | 52 | 25 | 1 | [9] |
| Freshwater fish | 71 | 16 | 1 | [23] |
| Tree frog | 564 | 33 | 1 | [18] |

* Isolation by the CMR leads to reciprocal monophyly (1), significant genetic structure (2), or no genetic differentiation (3) between eastern and western populations. $ The same species, *P. mandarinus*, utilized in this study.

In this study, we aimed to test the effect of sampling effort, specifically focusing on the sample size of individuals from eastern and western populations separated by the CMR, on the resulting mitochondrial phylogeography. Note that we understand that mitochondrial phylogeography can be erroneous because of, for example, the existence of nuclear copies of mitochondrial DNA (NUMTs), which has been recently identified in Odonata [24]. However, and while we fully acknowledge the limit of mitochondrial phylogeography [25,26] and the benefit of multilocus data and coalescent-based analyses for statistic phylogeography [2,5], mitochondrial gene genealogy is still, if not predominant, included in the majority of phylogeographic studies in animals. Specifically, mitochondrial phylogeography is often the first dataset that can be obtained to form testable hypothesis

and can be readily compared across multiple co-distributed taxa given the cornucopia of published data [3]. Furthermore, molecular-based species delimitation and the identification of cryptic genetic groups/species both rely heavily on mitochondrial datasets [27]. By assessing the effect of sampling effort on the resulting inferences based on the pattern of mitochondrial gene topology and population structure, our results will have broader impacts on not only phylogeographic studies *per se*, but also on how consistent the different types of biological entities that are identified as distinct genetic clusters are in molecular systematics that involve different sample sizes.

The endemic damselfly *Psolodesmus mandarinus* of Taiwan is a common and large-sized odonate that can be found close to creeks and small streams from low to mid-elevations in the mountain regions. There are three subspecies in Taiwan and the nearby Yaeyama islands, identified based on wing color patterns [28,29]. The Yaeyama subspecies, *P. m. kuroiwae*, is genetically distinct and divergent from the other two Taiwanese subspecies and has been elevated to full species status [16,30]. The two Taiwanese morphological subspecies are geographically structured to the northern, southern and eastern parts of Taiwan (Figure 1). However, intermediate forms can often be found. Unsurprisingly, the two Taiwanese subspecies did not form monophyletic mitochondrial groups in a previous study [16]; instead, the mitochondrial gene tree revealed two geographic lineages separated by the CMR [16]. However, one population from the east, Tongmen, has individuals from both the eastern and western lineages. The sampling from eastern Taiwan was limited in the previous study, and thus the extent of the geographic distribution of the two genetic lineages and the phylogeographic history of the species may not be correctly inferred. In this study, we expanded the geographic taxon sampling (a total of 124 localities; Figure 2 and Table 2) and increased the length of the sequenced mitochondrial region (a total of three mitochondrial loci; 1959 bp long) to study the mitochondrial genetic diversity and the geographic distribution of the genetic diversity. Specifically, we tested (1) whether the observed pattern of geographic lineages can be an artifact of limited sampling, (2) the effect of limited sampling on demographic inferences, and (3) based on our new data, we discuss the phylogeographic history of *P. mandarinus*.

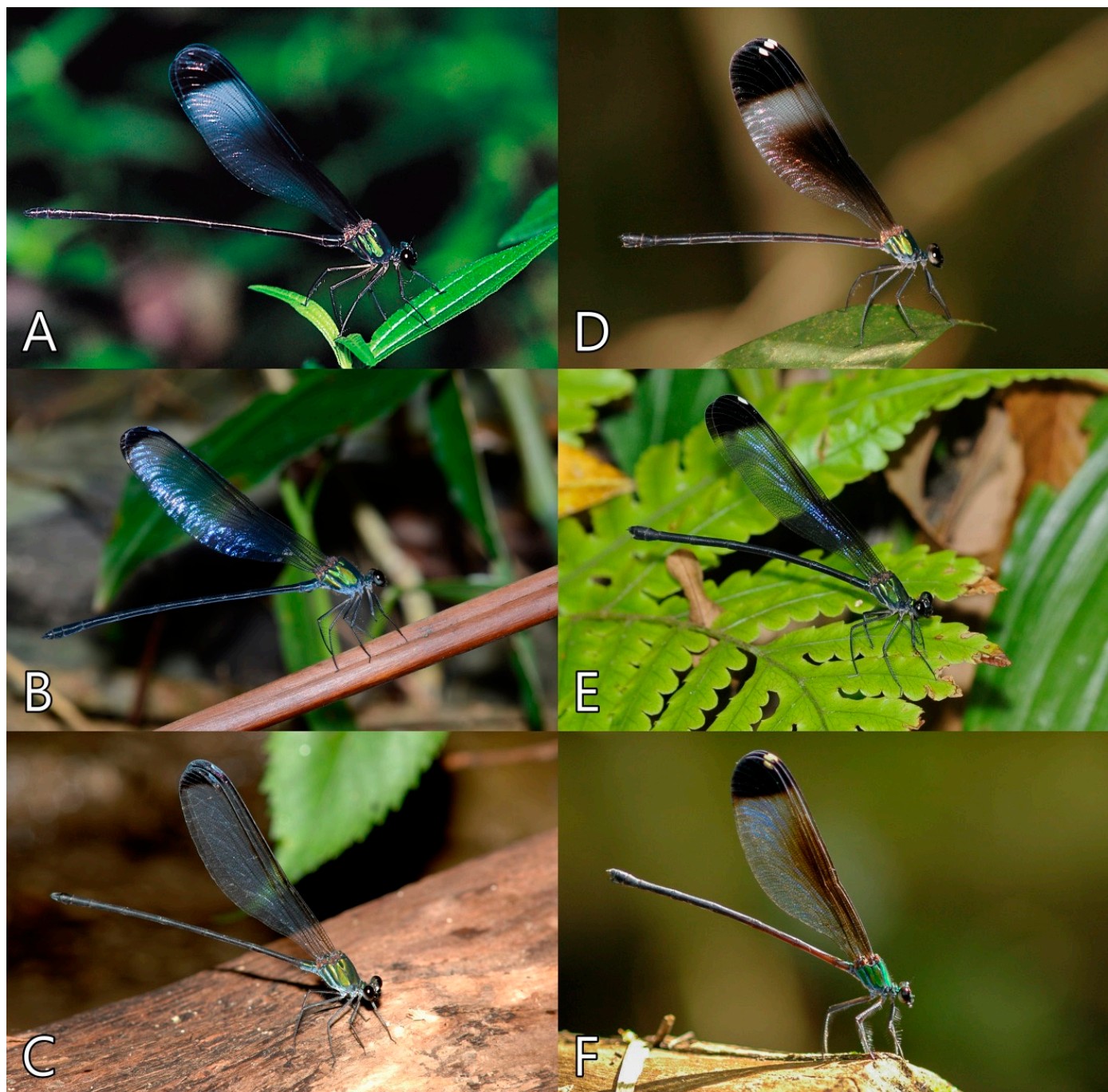

**Figure 1.** Populations of *Psolodesmus mandarinus* in Taiwan. (**A**–**C**) males. (**D**–**F**) females. (**A**) North Taiwan. Pamierh Park, Shihlin District, Taipei city. (**B**) South Taiwan. Shanping, Liukuei District, Kaohsiung city. (**C**) East Taiwan. Hsiama, Haituan Township, Taitung County. (**D**) North Taiwan. Wulai, Wulai District, New Taipei city. (**E**) South Taiwan. Neiwen, Neiwen Township, Pintung County. (**F**) East Taiwan. Fenglin, Fenglin Township, Hualien County.

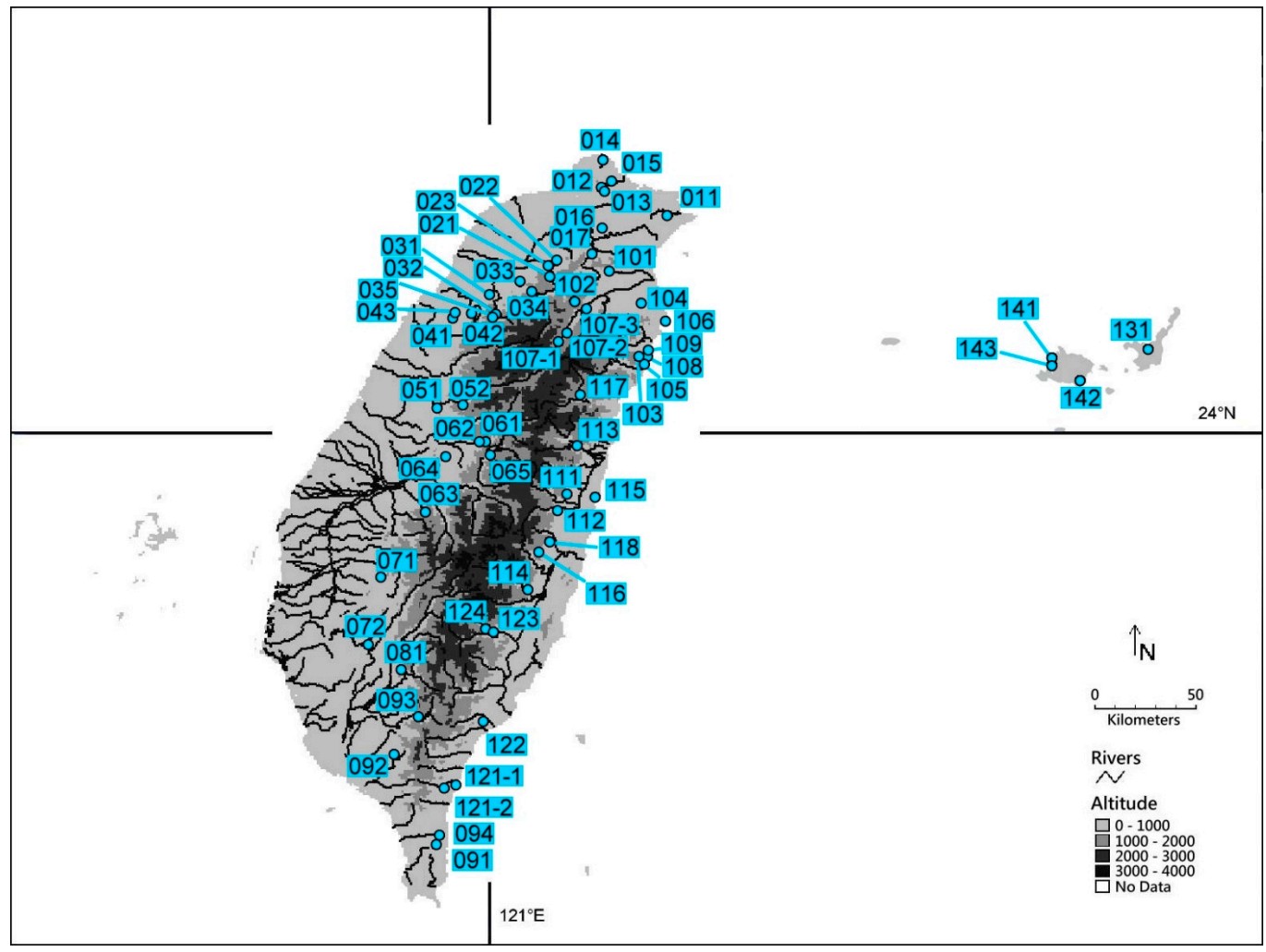

**Figure 2.** Distribution of sampling locations of *P. madarinus* in Taiwan and *P. kuroiwae* in Japan. Sampling locations are marked with open circles. The numbers of sampling locations are the same as in Table 2. The map was made using DIVA-GIS (https://www.diva-gis.org/; accessed on 1 June 2016).

**Table 2.** Sampling localities and their haplotype information.

| No. | Acronym | Locality | GPS Coordinates | Altitude | Haplotype • | Accession Numbers |
|---|---|---|---|---|---|---|
| Taiwan | | | | | | |
| 11 | TI | Tinglanku, Shuanghsi District, New Taipei city | 25°01′04.4″ N 121°52′32.0″ E | 42 m | H01 (2) | KM360534 |
| 12 | PI | Pingtenli, Shihlin District, Taipei city | 25°08′24.6″ N 121°34′43.1″ E | 500 m | H02 (1) | KM360535 |
| 13 | PA | Pamierh Park, Shihlin District, Taipei city | 25°07′20.6″ N 121°35′35.5″ E | 330 m | H01 (2) | KM360534 |
| 14 | AL | Alipang, Shihmen District, New Taipei city | 25°15′50.5″ N 121°35′05.2″ E | 140 m | H01 (1), H03 (2) | KM360534, KM360536 |
| 15 | LU | Lukuping, Wanli District, New Taipei city | 25°10′07.9″ N 121°37′18.3″ E | 419 m | H01 (4) | KM360534 |
| 16 | YI | Yinhotung, Hsintien District, New Taipei city | 24°57′30.5″ N 121°34′55.9″ E | 212 m | H01 (1), H04 (1) | KM360534, KM360537 |

**Table 2.** *Cont.*

| No. | Acronym | Locality | GPS Coordinates | Altitude | Haplotype • | Accession Numbers |
|---|---|---|---|---|---|---|
| 17 | WU | Wulai, Wulai District, New Taipei city | 24°50′20.6″ N 121°32′08.4″ E | 219 m | H01 (4) | KM360534 |
| 21 | JU | Junghua, Fuhsing Township, Taoyuan County | 24°44′05.5″ N 121°21′02.1″ E | 505 m | H01 (1), H05 (1) | KM360534, KM360538 |
| 22 | LI | Liuhsia, Fuhsing Township, Taoyuan County | 24°48′34.3″ N 121°22′25.5″ E | 364 m | H01 (2), H06 (1) | KM360534, KM360539 |
| 23 | FU | Fuhsing, Fuhsing Township, Taoyuan County | 24°47′20.9″ N 121°20′22.8″ E | 369 m | H07 (3) | KM360540 |
| 31 | PE | Peipu, Peipu Township, Hsinchu County | 24°39′27.8″ N 121°04′45.5″ E | 264 m | H07 (4) | KM360540 |
| 32 | SH | Shihlu, Chienshih Township, Hsinchu County | 24°33′58.6″ N 121°06′23.7″ E | 1110 m | H07 (2) | KM360540 |
| 33 | CS | Chienshihhsienho, Chienshih Township, Hsinchu County | 24°42′48.7″ N 121°12′32.4″ E | 300 m | H07 (1), H08 (1), H09 (1) | KM360540-KM360542 |
| 34 | CH | Chienshih, Chienshih Township, Hsinchu County | 24°40′11.1″ N 121°15′57.7″ E | 851 m | H01 (1), H07 (1), H10 (1) | KM360534, KM360540, KM360543 |
| 35 | KU | Kuanwu, Wufeng Township, Hsinchu County | 24°33′48.3″ N 121°05′35.8″ E | 812 m | H07 (2), H09 (1) | KM360540, KM360542 |
| 41 | ST | Shihtanpeitawo, Shihtan Township, Miaoli County | 24°33′09.5″ N 120°54′58.5″ E | 272 m | H07 (1) | KM360540 |
| 42 | NA | Nanchuang, Nanchuang Township, Miaoli County | 24°34′16.9″ N 121°00′00.1″ E | 332 m | H07 (4) | KM360540 |
| 43 | TO | Touwu, Touwu Township, Miaoli County | 24°34′40.8″ N 120°55′34.3″ E | 179 m | H07 (1), H08 (1), H11 (1) | KM360540, KM360541, KM360544 |
| 51 | HS | Hsinshe, Hsinshe District, Taichung city | 24°08′54.9″ N 120°50′38.7″ E | 605 m | H07 (3), H12 (1) | KM360540, KM360545 |
| 52 | KK | Kukuan, Hoping District, Taichung city | 24°09′28.3″ N 120°57′38.9″ E | 687 m | H07 (2), H13 (1) | KM360540, KM360546 |
| 61 | PP | Penpusi, Puli Township, Nantou County | 23°59′41.4″ N 121°03′41.1″ E | 735 m | H14 (2) | KM360547 |
| 62 | KY | Kuanyinpupu, Puli Township, Nantou County | 23°59′32.9″ N 121°02′06.0″ E | 646 m | H07 (1), H17 (1) | KM360540, KM360550 |
| 63 | HT | Hsitou, Luku Township, Nantou County | 23°40′27.8″ N 120°47′26.9″ E | 1082 m | H07 (7), H11 (1), H16 (1) | KM360540, KM360544, KM360549 |
| 64 | LH | Lienhuachih, Yuchih Township, Nantou County | 23°55′26.1″ N 120°53′03.5″ E | 735 m | H07 (2), H15 (1) | KM360540, KM360548 |
| 65 | JE | Jenai, Jenai Township, Nantou County | 23°55′42.4″ N 121°04′56.1″ E | 1120 m | H07 (2) | KM360540 |
| 71 | CP | Chungpu, Chungpu Township, Chiayi County | 23°23′13.2″ N 120°35′34.1″ E | 816 m | H07 (3) | KM360540 |
| 72 | NH | Nanhua Dam, Nanhua District, Tainan city | 23°04′38.3″ N 120°32′03.5″ E | 198 m | H07 (3) | KM360540 |
| 81 | SP | Shanping, Liukuei District, Kaohsiung city | 22°58′00.1″ N 120°41′02.5″ E | 660 m | H07 (4) | KM360540 |
| 91 | MU | Mutan, Mutan Township, Pintung County | 22°10′45.5″ N 120°50′26.5″ E | 280 m | H18 (3) | KM360551 |

**Table 2.** *Cont.*

| No. | Acronym | Locality | GPS Coordinates | Altitude | Haplotype $^\bullet$ | Accession Numbers |
|---|---|---|---|---|---|---|
| 92 | TA | Taiwu, Taiwu Township, Pintung County | 22°35′11.8″ N 120°38′55.3″ E | 395 m | H19 (3), H20 (2), H21 (1) | KM360552-KM360554 |
| 93 | WT | Wutai, Wutai Township, Pintung County | 22°45′22.8″ N 120°45′34.2″ E | 438 m | H07 (2) | KM360540 |
| 94 | NE | Neiwen, Neiwen Township, Pintung County | 22°13′24.4″ N 120°51′22.1″ E | 321 m | H18 (3) | KM360551 |
| 101 | TP | Tsaopi, Yuanshan Township, Yilan County | 24°45′41.4″ N 121°36′42.6″ E | 603 m | H01 (1) | KM360534 |
| 102 | MI | Mingchih, Tatung Township, Yilan County | 24°37′54.6″ N 121°27′11.7″ E | 1047 m | H01 (2) | KM360534 |
| 103 | SM | Shenmihu, Nanao Township, Yilan County | 24°22′41.3″ N 121°44′48.8″ E | 1100 m | H01 (1) | KM360534 |
| 104 | TU | Sanfu, Tungshan Township, Yilan County | 24°37′03.1″ N 121°45′23.9″ E | 140 m | H01 (2) | KM360534 |
| 105 | KF | Kufeng, Nanao Township, Yilan County | 24°20′41.0″ N 121°46′15.7″ E | 18 m | H01 (3), H22 (1) | KM360534, KM360555 |
| 106 | SU | Suao, Nanao Township, Yilan County | 24°32′18.6″ N 121°51′55.4″ E | 314 m | H01 (3) | KM360534 |
| 107-1 | SE | Province Highway 7A, Nanao Township, Yilan County | 24°26′41.3″ N 121°23′02.5″ E | 1088 m | H26 (1) | KM360559 |
| 107-2 | SE | Province Highway 7A, Nanao Township, Yilan County | 24°29′09.7″ N 121°25′30.5″ E | 781 m | H01 (1), H26 (1) | KM360534, KM360559 |
| 107-3 | SE | Province Highway 7A, Nanao Township, Yilan County | 24°35′37.7″ N 121°30′32.5″ E | 355 m | H01 (1) | KM360534 |
| 108 | NN2 | Nanao II, Nanao Township, Yilan County | 24°22′57.3″ N 121°47′02.0″ E | 220 m | H01 (2), H22 (1), H23 (1), H24 (1), H25 (1) | KM360534, KM360555-KM360558 |
| 109 | NN1 | Nanao I, Nanao Township, Yilan County | 24°24′03.2″ N 121°47′09.7″ E | 190 m | H01 (1), H23 (1), H27 (1) | KM360534, KM360556, KM360560 |
| 111 | FE | Fenglin, Fenglin Township, Hualien County | 23°45′29.9″ N 121°25′23.6″ E | 249 m | H32 (3), H37 (1), H38 (1), H40 (1), H41 (1), H47 (1), H50 (1), H51 (1), H55 (1) | KM360565, KM360570, KM360571, KM360573, KM360574, KM360580, KM360583, KM360584, KM360588 |
| 112 | KL | Kuangfulintao, Wanjung Township, Hualien County | 23°40′57.0″ N 121°22′58.1″ E | 229 m | H39 (1), H45 (1), H53 (1), H54 (1), H56 (1) | KM360572, KM360578, KM360586, KM360587, KM360589 |
| 113 | TM | Tungmen, Hsiulin Township, Hualien County | 23°58′39.4″ N 121°28′22.0″ E | 198 m | H28 (1), H29 (1) | KM360561, KM360562 |
| 114 | NNN | Nanan, Chohsi Township, Hualien County | 23°19′35.7″ N 121°14′26.3″ E | 445 m | H30 (3), H31 (1), H32 (1) | KM360563-KM360565 |
| 115 | CY | Chienying, Fenglin Township, Hualien County | 23°44′52.6″ N 121°32′53.9″ E | 160 m | H43 (1), H44(1) | KM360576, KM360577 |
| 116 | JS | Juisui, JuiSui Township, Hualien County | 23°29′45.0″ N 121°17′43.8″ E | 1141 m | H41 (2), H42 (1), H43 (3), H46 (1), H48 (1) | KM360574-KM360576, KM360579, KM360581 |

**Table 2.** *Cont.*

| No. | Acronym | Locality | GPS Coordinates | Altitude | Haplotype ● | Accession Numbers |
|---|---|---|---|---|---|---|
| 117 | HP | Hsipao, Hsiulin Township, Hualien County | 24°12′26.3″ N 121°28′54.6″ E | 939 m | H33 (1), H34 (1), H35 (1), H36 (1) | KM360566-KM360569 |
| 118 | FY | Fuyuan, JuiSui Township, Hualien County | 23°32′40.7″ N 121°20′37.1″ E | 898 m | H43 (1), H45 (1), H49 (1), H52 (1) | KM360576, KM360578, KM360582, KM360585 |
| 121-1 | TT | Tachu Main Stream, Tawu Township, Taitung County | 22°25′59.2″ N 120°52′45.4″ E | 288 m | H18 (3), H67 (1) | KM360551, KM360600 |
| 121-2 | TTB | Tachu Tributary, Tawu Township, Taitung County | 22°26′59.0″ N 120°55′46.1″ E | 123 m | H18 (3) | KM360551 |
| 122 | CI | Chihpen, Peinan Township, Taitung County | 22°44′09.2″ N 121°03′00.8″ E | 137 m | H68 (1) | KM360601 |
| 123 | TY | Tsiayunchiao, Haituan Township, Taitung County | 23°08′21.1″ N 121°05′59.8″ E | 475 m | H69 (1), H70 (1) | KM360602, KM360603 |
| 124 | HM | Hsiama, Haituan Township, Taitung County | 23°09′09.3″ N 121°03′53.7″ E | 680 m | H57 (1), H58 (1), H59 (1), H60 (1), H61 (1), H62 (1), H63 (1), H64 (1), H65 (1), H66 (1) | KM360590-KM360599 |
| **Japan** | | | | | | |
| 131 | OM | Ishigaki, Mt.Omoto | 24°25′15″ N 124°11′02″ E | 300 m | H72 (3), H73 (1), H74 (1), H75 (1), H76 (1), H77 (1) | H75: KM360604 |
| 141 | SO | Iriomote, Sonai | 24°23′17″ N 123°44′59″ E | 50 m | H78 (1), H82 (1), H85 (1) | H82: KM360606 |
| 142 | OH | Iriomotea, Otomi | 24°17′09″ N 123°52′55″ E | 80 m | H79 (1), H81 (2), H83 (1), H84 (1) | H79: KM360605, H84: KM360607 |
| 143 | SR | Iriomotea, Sirahama | 24°21′35″ N 123°45′06″ E | 60 m | H80 (1), H86 (1), H87 (1), H88 (1), H89 (1) | |

● Haplotypes were identified based on the concatenated sequences.

## 2. Materials and Methods

### 2.1. Sampling, DNA Extraction, Sequencing, and Alignment

Specimens of *Psolodesmus mandarinus* (186 specimens) were collected from 124 localities in Taiwan (Table 2, Figure 2), and *Psolodesmus kuroiwae* (21 specimens) were collected from four localities on Ishigaki and Iriomote Islands between 1999 and 2008. Most specimens were dried-preserved, and 2–5 legs for each specimen were removed and preserved in 95% ethanol for molecular studies. The voucher specimens were deposited in the Laboratory of Systematic Entomology and Forest Biodiversity, Taiwan Forestry Research Institute, Taipei, Taiwan. Total genomic DNA was extracted from one or two legs of each specimen using DNeasy Blood & Tissue Kits (Qiagen, Hilden, Germany) or the QuickExtract™ DNA Extraction Solution Kit (Epicentre, Madison, WI, USA) following the manufacturer's protocol. DNA samples were stored at −20 °C. We sequenced three mitochondrial loci (COI, tRNA-Leu, COII) with three sets of primers designed in this study (Table 3). Polymerace chain reactions(PCR) were carried out in a total volume of 25 μL, containing $10\times$ reaction buffer, 0.25 mM dNTPs, 2.0 mM MgCl2, 0.4 μM of each primer, 0.2 μL of Super-Therm polymerase (Hoffman-La-Roche, USA), 12.8 μL ddH$_2$O, and 3 μL of DNA template in an GeneAmp PCR System 9700 (Applied Biosystems, Foster City, CA, USA). The PCR profile was as follows: denaturing at 94 °C for 5 min, 35 cycles of amplification at 94 °C for 50 s followed by 50 °C for 50 s and 72 °C for 50 s, and a final extension at 72 °C for 7 min. PCR products were stained with ethidium bromide and visualized under UV light using

1.0% agarose gel after electrophoresis. PCR products were purified using a Gel/PCR DNA Fragments Extraction Kit (Geneaid, Taipei, Taiwan) and sequenced in both directions on an ABI PRISM™ 3730 automatic sequencer (Perkin Elmer, USA) at the Genomics BioSci & Tech, Taiwan. The three overlapping segments of amplified DNA were manually concatenated into a single sequence. Concatenated sequences were aligned without gaps using Clustal W [31]. We checked for pre-matured stop codons in the COI and COII sequences in order to identify possible NUMTs using Mega 6.0 [32]. Sequences generated in this study have been deposited in GenBank (KM360534-KM360607).

**Table 3.** Primers used in this study.

| Set Name | Primer Name | Primer Sequence (5′-3′) | Direction | Length | Amplification Region (Mt Gene) |
|---|---|---|---|---|---|
| Pmk-005 | Pmk-F001 (Pmk-COI-1684F) Pmk-R005 (Pmk-COI-2346R) | CCCACGACTAAACAACATAAG GGAACAGCAATTACTATTGTGG | forward reverse | 663 bp | COI |
| Pmk-006 | Pmk-F006 (Pmk-COI-2178F) Pmk-R006 (Pmk-COI-2917R) | CCCAAGAAAGAGGAAAGAAG GAATCTATGTTCTGTTGGTGG | forward reverse | 740 bp | COI |
| Pmk-007 | Pmk-F007 (Pmk-COI2895F) Pmk-R007 (Pmk-COI3708) | CACCACCAACAGAACATAG GTCATCTAGTGAGGCTTCAC | forward reverse | 814 bp | COI-tRNA-Leu-COII |

### 2.2. Phylogenetic and Network Analyses

The phylogenetic analyses of samples from *P. mandarinus* and the congener *P. kuroiwae* were performed using Maximum Likelihood (ML) and Bayesian Inference (BI). The best-fitting model was chosen as the model of molecular evolution in both the ML and the BI by the hierarchical likelihood ratio tests in jModeltest 2.1 [33]. ML phylogenetic reconstruction was conducted using Mega 6.0 [32], and ML branch supports were calculated with 1000 bootstrap replicates [34]. BI phylogeny was performed using MrBayes v.3.2 [35]. MCMC runs for 10 million generations were repeated twice, with trees sampled every 100 generations. The first 25,000 trees in each run were discarded as burn-in, and the remaining trees were used to construct Bayesian consensus trees. A statistical parsimony haplotype network was constructed using TCS v. 1.21 [36]. The maximum mutational step was set at 130 for connections between haplotypes. An additional run with a parsimony probability set at 0.95 was performed to test the statistical supports of connections. Gaps were treated as the fifth character state regardless of the length.

### 2.3. Population Genetic Analyses

Each individual sample was assigned to either the eastern or the western population according to hypothesized phylogeographic breaks (Figure 2). The southern regions of Taiwan may have phylogeographic affinities to either eastern or western Taiwan, which varies across studied organisms. Here, we assigned individuals from southern Taiwan to the western region in that studies using aquatic or riverine organisms often reveal a southwestern genetic clade (e.g., [18,37]). The summary statistic of population size (θ) estimated based on the number of segregating sites per site was calculated for the eastern and western populations separately using the theta.s function implemented in the R package pegas [38]. A neutrality test of the aligned sequences in different populations was performed using the Tajima's D index via the tajima.test function in pegas. Furthermore, genetic differentiation between populations was calculated using the diff_stats function in the mmod package [39]. Specifically, Nei's GST, Jost's D, and ΦST were calculated from our mitochondrial dataset.

### 2.4. Testing the Effect of Sampling Effort on Inferring Population Structure Phylogenetic Reconstruction

In order to test the effect of sampling effort on phylogeographic studies, we randomly sampled our sequences with different numbers of individuals to represent the eastern and western populations by a customized R script using R (https://www.r-project.org/;

accessed on 1 June 2016). Specifically, we randomly subsampled the eastern and western populations of our DNA sequences (a total of 300 subsampled sequence alignments) 100 hundred times for 10, 20, and 50 individuals. A phylogenetic tree was reconstructed for each alignment using the neighbor-joining method with a TN93 model with a gamma variable for rate correction via the dist.dna and nj functions rooted using the midpoint method implemented in the R package ape [40]. The reciprocal monophyly of eastern and western populations in each reconstructed tree was then assessed using the is.monophyletic function in ape. We then reported how often reciprocal monophyly was observed with different levels of sampling effort. Furthermore, the population size ($\theta$) of each population and genetic differentiation between populations ($\Phi$ST) were also calculated for each subsampled alignment. Whether different sampling efforts can result in significantly different values of population genetic parameters was tested using the 100 replicates of each dataset.

## 3. Results

### 3.1. Sequence Alignment, Phylogenetic, and Network Analyses

A total of 186 individuals were successfully sequenced; 86 of them from the western population and the remaining 100 from the eastern population. The sequence alignment was 1959 bp in length with 43 and 73 parsimony uninformative and informative sites, respectively. A total of 70 unique haplotypes were found in *P. mandarinus,* and 19 were identified in *P. kuroiwae*. Four haplotypes of *P. kuroiwae* were selected as outgroups for the phylogenetic analyses. The monophyly of *P. mandarinus* was supported in the phylogenetic analyses (Figure 3). The phylogeny reconstructed based on mitochondrial loci (COI-tRNA-Leu-COII) revealed that two distinct haplotype clades (widespread and East) exist in *P. mandarinus* in Taiwan, which were separated by a deep phylogenetic split. The widespread clade included the haplotypes distributed throughout Taiwan. The geographical distributions of two high-frequency haplotypes H01 (38 individuals) and H07 (48 individuals) ranged from North–East (Nanao, Yilan County) to North (Chienshih, Hsinchu County) and from North (Chienshih, Hsinchu County) to South (Shanping, Kaohsiung city) (Figure 4 & Table 2), respectively. The eastern clade that was restricted to eastern Taiwan ranged from southern Yilan County (No. 109, Nanao I) to northern Taitung County (No. 124, Hsiama). The eastern clade contained two subclades, one smaller subclade distributed in Nanao, southern Yilan County (three haplotypes: H22, H23, H25) and one larger subclade distributed in Hualien County to Hsiama, northern Taitung County. The site Hsiama (No. 124 in Haituan Township, Taitung County, Table 1) had 10 haplotypes, which were geographically widely distributed haplotypes that belong to most of the subclades within the two main clades.

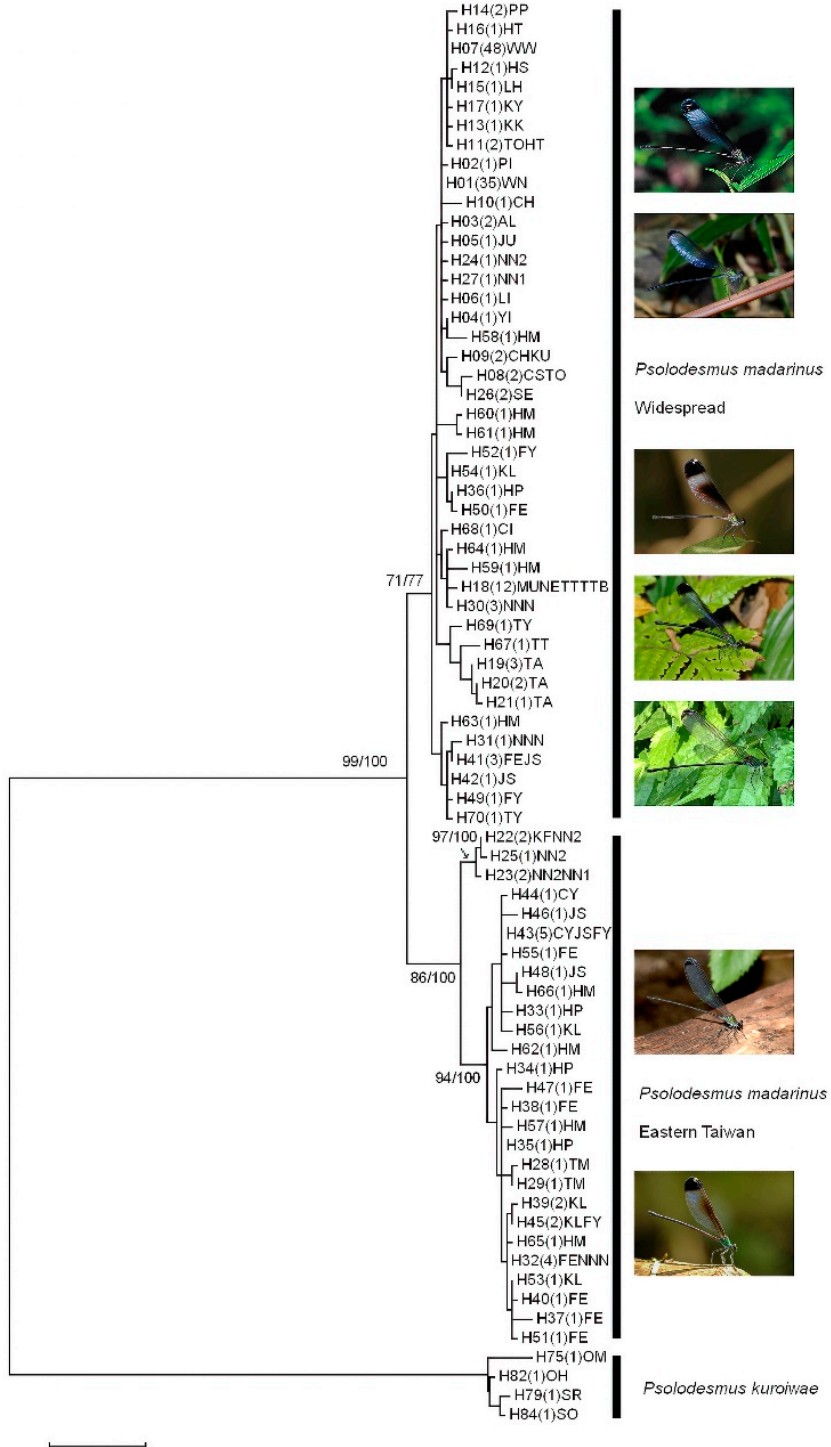

**Figure 3.** The phylogenetic tree of *Psolodesmus madarinus* haplotypes from the maximum likelihood analysis based on the HKY + G model. Bootstrap values from the maximum likelihood (ML) analyses together with the posterior probabilities from the Bayesian analysis (BI) are indicated (ML/BI) near the branches. Groups of haplotypes are labelled according to whether they are found widespread in Taiwan or endemic to eastern Taiwan.

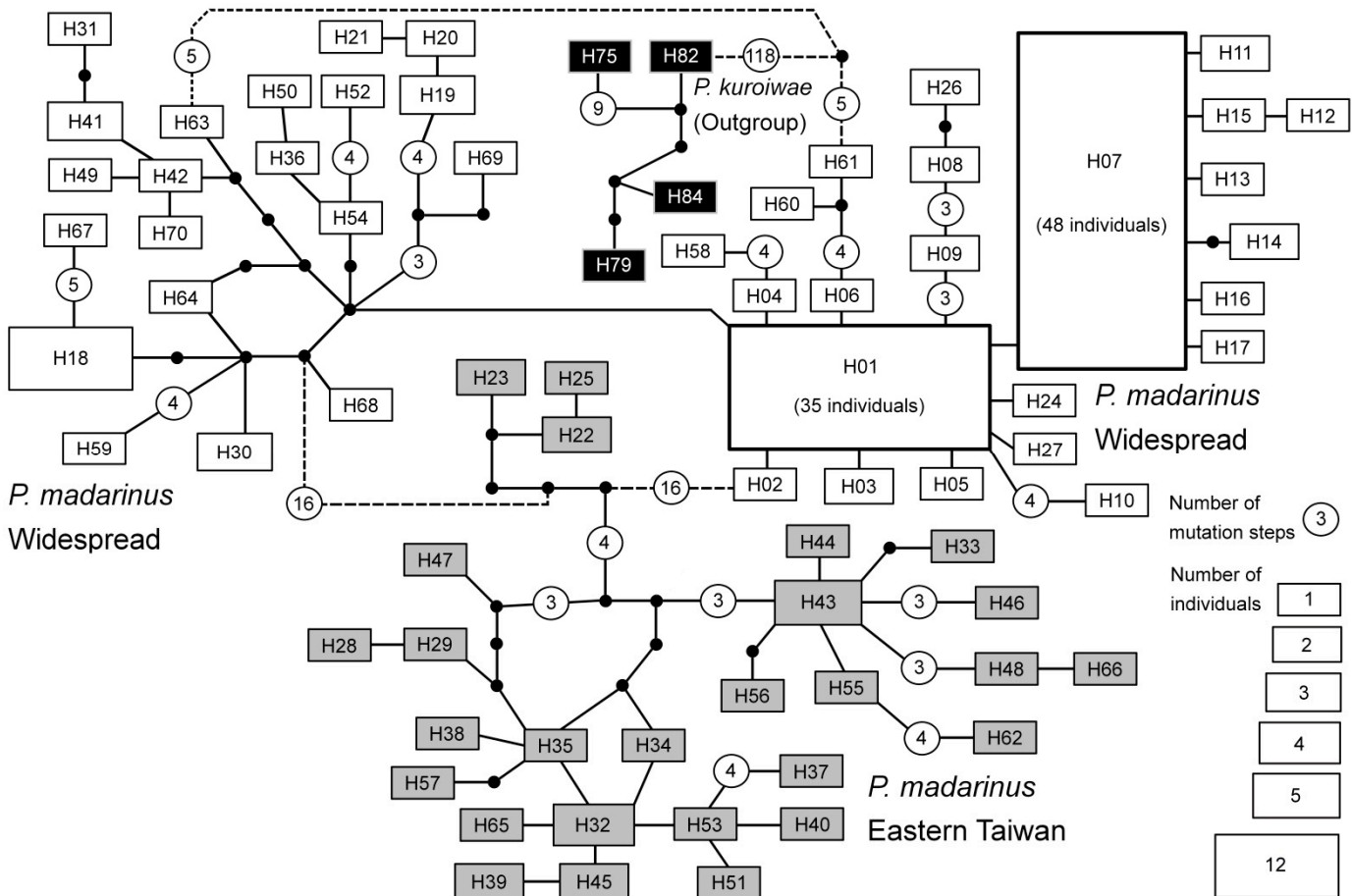

**Figure 4.** Statistical parsimony network of *P. madarinus* haplotypes. Haplotype groups that are widespread in Taiwan, endemic to eastern Taiwan or belong to *P. kuroiwae* (outgroup) are marked with white, grey and black rectangles, respectively. The sizes of the rectangles for each haplotype are proportional to the number of individuals found carrying each haplotype. Black dots represent hypothetical and unobserved intermediate haplotypes. Solid lines between haplotypes represent one mutational step. For haplotypes connected with more than three mutational steps, open circles with numbers are applied, indicating the number of mutational steps between haplotypes. Dashed lines represent connections between haplotypes with statistical probability < 95%.

*3.2. Population Genetic Analyses*

The estimated θs for the western and eastern populations were 4.775418 and 19.50795, respectively. The estimated Tajima's Ds were −2.15586 (*p* = 0.01) and −0.4803459 (*p* = 0.67), respectively; the *p* value assumes that D follows a beta distribution after rescaling on [0,1]. Therefore, the eastern population may have a larger estimated effective population size than the western population because of the higher estimated genetic diversity; on the other hand, only the western population may have experienced a recent population expansion because of a significantly negative Tajima's D value. The calculated Nei's GST, Jost's D, and ΦST from the dataset were 0.02, 0.81, and 0.85, respectively and the significance of population subdivision was estimated as 0.000999 based on 1000 permutations. That is, a statistically significant population structure between the western and the eastern populations was identified.

*3.3. The Effect of Sampling Effort on Phylogenetic Reconstruction and Inferring Population Structure*

Phylogenetic relationships reconstructed using the datasets that randomly selected 10, 20, and 50 individuals from western and eastern populations did not result in reciprocal monophyletic geographic groups (0 out of the 300 datasets). Furthermore, none of the datasets supported a monophyletic eastern lineage. Nevertheless, 8 out of the 100 datasets that randomly drew 10 individuals per population resulted in a monophyletic western lineage, while results from the datasets that randomly chose 20 and 50 individuals per population did not support a monophyletic western lineage (0 out of 200 datasets). Therefore, a monophyletic geographic lineage of *P. mandarinus* can result when a small sample size is used.

Most of the subsampled alignments led to smaller θ values than the original alignment for both western and eastern populations (99, 100, and 94 times for the western population and 99, 99, and 94 times for the eastern population from datasets using only 10, 20, and 50 individuals per population, respectively). On the other hand, the calculated Tajima's D values were very often higher for the subsampled alignments than for the original dataset (100, 100, and 82 times for the western population and 80, 92, and 93 times for the eastern population from datasets using only 10, 20, and 50 individuals per population, respectively). Three out of the 100 subsampled alignments containing 10 individuals per population resulted in significant ($p < 0.05$) negative values of Tajima's D for the western population, while only one showed significant negative values for the eastern population. For the 20 individuals per population dataset, 29 of the 100 datasets indicated significant negative Tajima's D values, while none resulted in significant negative D values for the western and eastern populations, respectively. Finally, 89 of the 100 datasets that subsampled 50 individuals per population resulted in significant negative Tajima's Ds for the western population, while none of them was significant for the eastern population. Hence, small sampling sizes in a phylogeographic study result in a smaller inferred effective population size and biased pattern of demographic expansion in *P. mandarinus* (Figure 5).

The estimated ΦST values between western and eastern populations were very often higher from the subsampled alignments than the original dataset (Figure 5). However, this pattern was more significant in datasets that contained more individuals per population. Specifically, the 10 individuals per population dataset resulted in 76 ΦST values (out of 100) larger than the estimate from the original dataset; the 20 individuals per population dataset resulted in 95 values that were larger than the original; finally, the ΦST values estimated from 50 individual datasets were all larger than that estimated from the original dataset. The datasets with a sample size of 10 individuals had the highest standard deviations (SDs = 0.096, 0.074, and 0.027 for 10, 20, and 50 individuals per population dataset, respectively), showing a wider range of estimate values, while similar mean values of ΦST were found among datasets (0.197, 0.214, and 0.214).

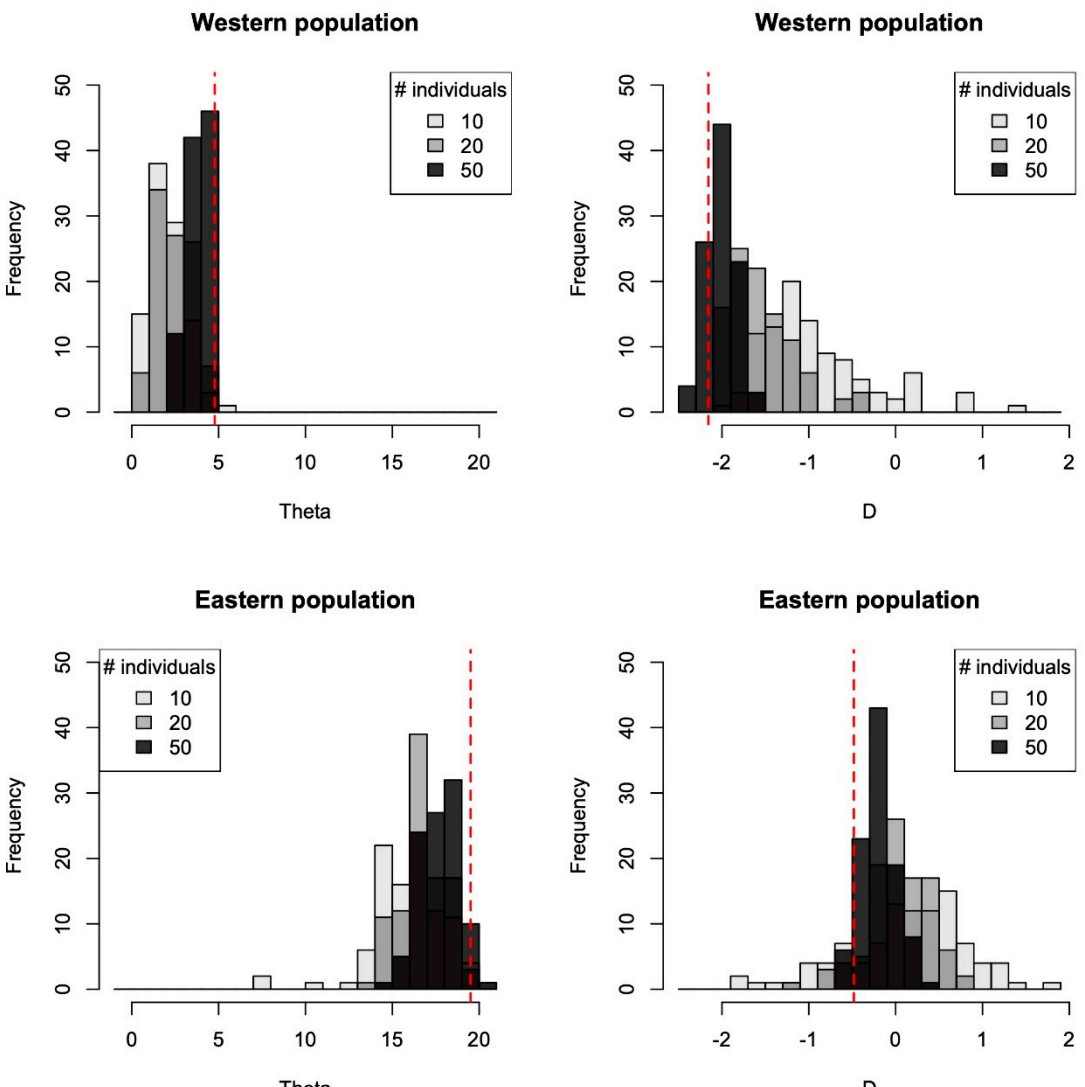

**Figure 5.** The values of θ and Tajima's D calculated for western and eastern populations of *Psolodesmus mandarinus*. The red dashed lines indicate the values estimated from the original dataset (86 individuals from western and 100 individuals from eastern populations).

## 4. Discussion

The sampling effect on the resulting evolutionary inferences has long been a point of argument in phylogeographic studies. However, few studies have explicitly tested the effect of different sampling intensities on the possible inferences using empirical data sets. We studied the phylogeographic history of *Psolodesmus mandarinus* using 186 samples from the entire geographic distribution with a mitochondrial locus containing 1959 sites. We found two distinct lineages from the phylogenetic and network analyses (Figures 3 and 4; c.f., [16]), where one of the lineages was geographically widespread while the other was restricted to the eastern part of the CMR. Although the eastern and western populations subdivision by the CMR does not lead to reciprocal monophyly in the *P. mandarinus* system, geographic genetic structure is apparent, indicating that the CMR does have a strong effect on isolating geographic populations from either side of the mountain range. We further showed that although reciprocal monophyly cannot be found from our complete dataset, by subsampling a subset of individuals from eastern and western populations, a monophyletic western lineage could sometimes be found. The estimated population genetic parameters could be biased when a subset of individuals was used in the analyses; for example, a significant demographic expansion inferred for the western population became

less apparent when only a subset of samples was included. Our results imply that different sampling efforts (specifically, the number of individuals per population) may in part explain different phylogeographic histories inferred among different empirical studies in Taiwan (Table 1). We discuss the ramifications of our findings on phylogeographic inferences and provide a revised phylogeographic history of *P. mandarinus* in the following sections.

### 4.1. The Effect of Sampling Effort on the Reconstructed Phylogeographic History

Our results clearly indicate that by subsampling our mitochondrial dataset, not only can the reconstructed phylogenetic inferences (i.e., whether a monophyletic geographic group can be observed) be impacted, but also the estimated population genetic parameters (Figures 5 and 6). Since the number of individuals sampled per geographic population/locality varies across phylogeographic studies (c.f., this study and [16]; see also Table 1), our results imply that different phylogeographic inferences made across different empirical studies may result from different sampling efforts. Many studies have attributed different phylogeographic patterns observed among systems to differences in species-specific ecological and biological properties, while an alternative hypothesis that such differences can result simply because of unequal sampling efforts has rarely been tested [1,2]. We have shown that different sampling efforts can indeed result in very different phylogeographic inferences—specifically, (1) whether a geographic population may appear to be monophyletic, (2) whether a population expansion event can be identified, and (3) whether a significant geographic genetic structure can be detected.

**Figure 6.** The fixation indices ($\Phi$ST) calculated between western and eastern populations. The red dashed line indicates the original estimated value (86 individuals from western and 100 individuals from eastern populations).

By reviewing selected phylogeographic studies in Taiwan (Table 1), we also found that studies that inferred the phylogeographic patterns of (1: reciprocal monophyly) and (3: no genetic divergence between population) often had a smaller sampling size (in terms of the number of individuals or the number of localities) than studies that resulted in pattern (2: apparent genetic divergence between geographic populations) (Figure 7). That is, without considering the biological and ecological differences among the selected empirical phylogeographic systems (Table 1), sampling size difference alone may explain at least in part the different phylogeographic patterns found in different empirical studies. Specifically, our various subsampling designs revealed that a small sampling size can lead to a higher probability of observing monophyletic geographic groups (phylogeographic pattern 1) and a less apparent geographic genetic structure (phylogeographic pattern 3). Note that we are not discrediting the importance of biological and ecological differences among evolutionary lineages that can lead to different phylogeographic patterns. It has been shown that with more than 500 sampled individuals and 33 localities, there was no significant geographic genetic structure in a treefrog species [37]. Our study instead goes against making direct links between the biological and ecological properties of the study system with the inferred phylogeographic pattern because the inferred phylogeographic pattern may be affected by many other stochastic factors [2], and sampling effort could be one such factor.

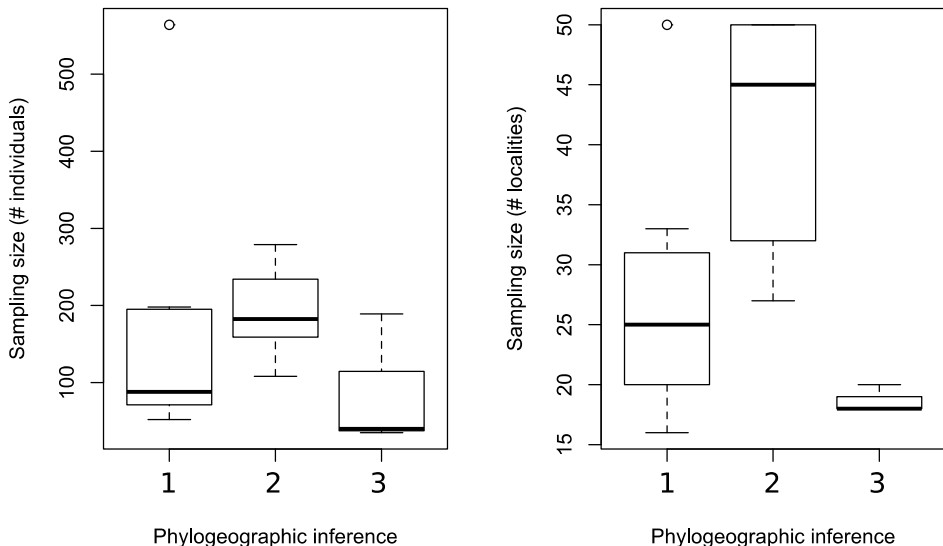

**Figure 7.** Summary of sampling sizes, i.e., number of individuals and number of localities, from selected empirical phylogeographic studies that resulted in different evolutionary inferences. 1. Studies that showed monophyletic geographic populations. 2. Studies that revealed significant genetic differentiation between geographic populations. 3. Studies that inferred panmictic geographic populations.

*4.2. The Phylogeographic History of Psolodesmus mandarinus*

Our results agree with the inferences made by a previous phylogeographic study on *P. mandarinus*: (1) the Yaeyama taxon forms a distinct evolutionary lineage, (2) there is an eastern Taiwan restricted lineage and a widespread lineage of the Taiwanese samples, and (3) the two Taiwanese lineages do not correspond to the subspecies assignment based on wing morphology [16]. However, by sequencing additional fragments from the mitochondrial genome and sampling more individuals and localities, we unraveled additional genetic diversity represented by the significant increase in the number of haplotypes (Figures 3 and 4 and Table 2). While only seven haplotypes were identified in [16], we have identified a total of 71 haplotypes. We further showed that there is a higher genetic diversity found in eastern Taiwan in addition to the fact that the eastern population harbors individuals from the two main evolutionary lineages. Our results therefore imply that eastern Taiwan is

likely the geographic origin of the Taiwanese *P. mandarinus*. The phylogeographic pattern and inference are in direct contrast to what has been hypothesized for another widespread damselfly species, *Euphaea formosa*, in Taiwan [14]. The genus *Psolodesmus* belongs to the family Calopterygidae, which is most abundant in temperate regions; on the other hand, the diversity center of the family Euphaeidae, to which the genus *Euphaea* belongs, is in the tropics [41]. Taiwan is a subtropical island that harbors species of both temperate and tropical origins that may have immigrated into Taiwan via different historical routes [42,43]. *P. mandarinus* and *E. formosa* might have colonized Taiwan via different historical routes because of their differences in ecological preferences and geographic origins.

While the eastern population may have a stable effective population size through recent history, a population size expansion was inferred for the western CMR population of *P. mandarinus*. The inferred contrasting demographic histories also imply that the western population was founded by the eastern population, where the recently founded population went through size expansion event after colonizing previously unoccupied habitats. On the other hand, the low to mid- elevation riverine habitats of western Taiwan cover a larger geographic area than those of eastern Taiwan (Figure 1). Furthermore, the river systems of western Taiwan might have been connected to form a much larger river system during periods of lowered sea level [14]. An increase in the population size of the western population may thus also be impacted by geo-historical events. While we could not effectively test which factor may have played a major role in the demographic history of *P. mandarinus*, it is likely that both evolutionary history and geo-historical events were involved in shaping the population genetic diversity and divergence as shown in other Taiwanese taxa (e.g., [8–10,14]).

## 5. Conclusions

Phylogeographic studies depend on the sampling design, e.g., the number of geographic populations and the number of individuals per population, to understand the spatial variation of intraspecific genetic diversity and to make inferences regarding the origin and maintenance of biodiversity. We showed that different sampling designs can impact the pattern revealed by the genetic data and thus lead to different inferences regarding the effect of a geographic barrier. We further demonstrated that different phylogeographic patterns observed among biological systems in Taiwan, although often being attributed to their biological differences, may simply be the result of different geographic sampling strategies. We argue that future phylogeographic studies require not only the careful design of spatial sampling strategies but also the testing of sampling effects on the resulting inferences as we have shown in our study.

**Author Contributions:** Conceptualization, L.-J.W. and J.-P.H.; methodology, L.-J.W., J.-P.H. and Y.-W.C.; data analysis, L.-J.W., J.-P.H. and Y.-W.C.; investigation, L.-J.W.; resources, L.-J.W.; data curation, L.-J.W., Y.-W.C. and J.-P.H.; writing—original draft preparation, L.-J.W. and J.-P.H.; writing—review and editing, L.-J.W., J.-P.H. and Y.-W.C. All authors have read and agreed to the published version of the manuscript.

**Funding:** This study was partially supported by the granted project No. 103AS-13.3.4.-FI-G2 from the Taiwan Forestry Research Institute.

**Institutional Review Board Statement:** Not applicable.

**Informed Consent Statement:** Not applicable.

**Data Availability Statement:** The sequence data presented in this study are openly available in GenBank of NCBI (National Center for Biotechnology Information) at [https://www.ncbi.nlm.nih.gov (accessed on 1 June 2016)]. See Table 2 for accession numbers.

**Acknowledgments:** We are grateful to Shih-Chieh Huang, Jyh-Jong Cherng, Hsien-Chih Wu, Ye-Zhi Ai, Chen-Hsiang Chen and Yung-Jen Chang for their help in the fieldwork. Two reviewers provided helpful suggestions that improved the clarity of the manuscript.

**Conflicts of Interest:** The authors declare no conflict of interest.

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
