# Peer review of "Testing the Effect of Sampling Effort on Inferring Phylogeographic History in Psolodesmus mandarinus (Calopterygidae, Odonata)"

_diversity, doi:10.3390/d14100809_

Round 1

Reviewer 1 Report

The paper analyzed is very interesting and valuable. It brings new data to the knowledge about the examined species, and raises, on its example, important problems generally related to phylogeographic research. I have no methodological or substantive objections, and the research model has been chosen perfectly. The presentation of data and its analysis on the background of the literature on the subject also raises no objections. The conclusions are of great importance for future research in the field covered by this article.

It is very good that the authors have studied several different mitochondrial loci. A recent study by Ozana et al. (2022) showed that nonfunctional nuclear copies of mitochondrial DNA can have a serious impact on barcoding, phylogenetic, population and phylogeographic studies of Odonata:

https://resjournals.onlinelibrary.wiley.com/doi/abs/10.1111/syen.12550

This also applies to some of the most frequently used sequences, incorrect data even ends up in public databases. The use of several various sequences reduces this risk. The authors cite numerous previous studies from Taiwan – I wonder what sequences were included in them? Perhaps the credibility of the data also depends on this.

In valuable conclusions, I miss more specific information that could directly affect the practice of research. Could the authors at least initially specify how large the material for phylogeographic research should be? Are there any minimum numbers of specimens and samples? In absolute numbers or in relation to e.g. the size of the area of distribution of a studied species? Such information would be of great methodological importance, allowing for better standardization of future research. Of course, it may be too early for that, but I'd like to even read assumptions or speculations.

Most of the few minor technical faults I marked in the pdf file. I found a few typing errors. The authors did not use italics in several places. In References, one journal name is given abbreviated, while full names are given elsewhere. The first 5 references are indented less than the rest. There is a space before references nos 17-20. Page 30 is blank.

I also pay attention to the fact that the words used in the title are repeated in Keywords. I do not know what the editorial policy is in this regard, but it should usually be avoided.

Author Response

  1. thank you for the suggested study of reference about NUMTs. We added a couple brief discussion in Introduction and method sections about NUMTs and we made sure that our results would not be significantly affected by NUMTs.
  2. In most of the case studies listed in our table 1, researchers utilized co1, co2, or control region for their mitochondrial phylogeographic studies. However, we believe that most of the studies translated their mtDNA sequences to check for stop codons (except for control region), as what we have done in our study.
  3. Thank you for the suggestion about our conclusion. We are however hesitate to make such broad/general argument about suitable sampling size for phylogeographic study. We believe to make such generalization a large scale simulation study considering different levels of isolation, population size, and geographic ranges of the simulated organisms is required. Our result on the other hand only serve as a cautionary tale to not make strong evolutionary inference when the sample size is not very big.

Reviewer 2 Report

I found this a very interesting paper exploring an important question in phylogeographic studies, namely the influence of sample size on model results.  I think the work is well-executed, and that the results are interpreted and presented appropriately.  I have identified a number of minor language corrections (please see attached pdf), but I think once these are addressed the manuscript is suitable for publication.

Author Response

thank you. We made changes in the manuscript according to your edits.